# Short-course radiotherapy combined with chemotherapy and PD-1 inhibitor in low-lying early rectal cancer: study protocol for a single-arm, multicentre, prospective, phase II trial (TORCH-E)

Yajie Chen [1,2] Yaqi Wang,[1,2] Hui Zhang [1,2] Juefeng Wan [1,2] Lijun Shen,[1,2] Yan Wang,[1,2] Menglong Zhou,[1,2] Ruiyan Wu,[1,2] Wang Yang,[1] Shujuan Zhou,[1] Sanjun Cai,[2,3] Xinxiang Li,[2,3] Zhen Zhang,[1,2] Fan Xia[1,2]

For numbered affiliations see end of article.

**Correspondence to**
Dr Fan Xia;
tcxiafan@hotmail.com,
Dr Zhen Zhang;
zhen_zhang@fudan.edu.cn and
Xinxiang Li;
xinxiangli@fudan.edu.cn

## ABSTRACT

**Introduction** Current standard treatment for patients with early rectal cancer is radical surgical resection. Although radical surgery provides effective local tumour control, it also increases the mortality risk and considerable adverse effects, including bowel, bladder, sexual dysfunction and loss of anal function, especially in patients with low-lying rectal cancer. Recent studies have shown promising synergistic effects of the combination of programmed cell death-1 (PD-1)/programmed cell death-ligand 1 (PD-L1) inhibitors and radiotherapy in improving tumour regression. For patients who reach a clinical complete response (cCR) after neoadjuvant therapy, a 'Watch & Wait' (W&W) approach can be adopted to preserve anorectal function and improve quality of life. Thus, this study aims to explore the efficacy and safety of radiotherapy combined with chemotherapy and PD-1 antibody in patients with low early rectal cancer.

**Methods and analysis** TORCH-E study is designed as a multicentre, prospective, phase II trial of short-course radiotherapy (SCRT) combined with chemotherapy and PD-1 inhibitor in patients with cT1-3bN0M0 low rectal cancer. The trial was initiated in December 2022 and is currently recruiting patients, with an anticipated completion of participant enrolment by June of the following year. The enrolled 34 patients will receive SCRT (25 Gy/5 Fx), followed by four cycles of capecitabine plus oxaliplatin chemotherapy and PD-1 antibody (toripalimab) and finally receive surgery or the W&W strategy. The primary endpoint is the complete response (CR) rate, that is, the rate of pathological complete response (pCR) plus cCR. The secondary endpoints include organ preservation rate, 3-year local recurrence-free survival rate, 3-year disease-free survival rate, 3-year overall survival rate, grade 3–4 adverse effects rate and patients' quality of life.

**Ethics and dissemination** This trial has been approved by the Ethics Committee of Fudan University Shanghai Cancer Center. Trial results will be disseminated via peer-reviewed journals and conference presentations.

**Trial registration number** NCT05555888 (ClinicalTrials.gov).

## STRENGTHS AND LIMITATIONS OF THIS STUDY

⇒ This clinical trial is the first prospective, multicentre, phase II study using short-course radiotherapy combined with programmed cell death-1 (PD-1) inhibitor and chemotherapy in early low-lying rectal cancer.

⇒ This study aims to investigate the impact of short-course radiotherapy on tumour regression and its synergy with immunotherapy, differing from previous studies employing long-course radiotherapy.

⇒ The participants with early low rectal cancer can attain non-operative management or minimally invasive surgery via a combination of radiotherapy, chemotherapy and PD-1 inhibitor, aiming for optimal organ function preservation.

⇒ Limitation: this is a phase II clinical study involving a small number of patients, a phase III trial with a larger sample size will be necessary to validate the effectiveness and safety.

## INTRODUCTION

The standard treatment for cT1-3bN0M0 (stage I–II) rectal cancer remains radical surgical resection. Despite effective local tumour control, surgical resection can elevate mortality risk and lead to notable adverse effects, such as high stoma rates, faecal incontinence and urinary and sexual dysfunction,[1] particularly in cases of low-lying rectal cancers. Most survivors encounter functional sequelae due to removal of the rectum, which significantly impairs their quality of life (QoL).

Neoadjuvant chemoradiotherapy (nCRT) is recognised for its potential to promote tumour regression, reduce local recurrence rates (LRTs) and enhance sphincter preservation in locally advanced rectal cancer (LARC). Patients who reach a complete clinical response (cCR) after nCRT can adopt the 'Watch & Wait' (W&W) strategy to preserve

BMJ

the rectum,[2] and the total neoadjuvant therapy (TNT) approach is also expected to decrease distant metastasis and improve long-term survival.[3 4] The evident benefits of nCRT in LARC have spurred interest in its application for early rectal cancer. Initial retrospective studies indicated comparable local tumour control between nCRT followed by transanal endoscopic microsurgery (TEM) and direct total mesorectal excision (TME) surgery for T2–3 rectal cancer.[5 6] A subsequent prospective multi-institutional phase II ACOSOG Z6041 study validated this finding, showing favourable 3-year disease-free survival, organ preservation rates and substantial tumour downstaging in patients with T2N0 low-stage rectal cancer who underwent local resection following nCRT.[7] These results suggest the potential of nCRT as an organ-preserving alternative for patients with low early rectal cancer, particularly those encountering challenges in anus preservation.

In recent years, numerous clinical studies have extensively explored the nCRT treatment approach for early rectal cancer. The Spanish TAUTEM study compared preoperative CRT and TEM with conventional TME surgery in T2–T3s (superficial) N0, M0 rectal cancer, and found that combining nCRT with TEM achieved a high pathological complete response (pCR) rate (44.3%).[8] Similarly, the STAR-TREC phase II trial adopted patients with mrT1-3b N0 M0, ≤40 mm tumour diameter. Randomised into TME, organ preservation followed by short-course radiotherapy (SCRT) or organ preservation followed by long-course chemoradiotherapy (LCRT) groups, the trial showed that nCRT was well-tolerated. The 24-month LRT was comparable to the TME group, with 60% achieving organ preservation.[9] The TAUTEM and STAR-TREC studies collectively demonstrate the viability of CRT combined with local resection to achieve optimal organ preservation in early rectal cancer. Additionally, the OPERA trial and WW2 study validate high-dose CRT's efficacy and safety, enabling patients with early rectal cancer to attain radical outcomes via chemoradiation, thereby preserving the rectum.[10 11]

Preclinical evidence suggests that combining radiotherapy with programmed cell death-1 (PD-1)/programmed cell death-ligand 1 (PD-L1) antibodies can reshape the tumour microenvironment, counter immunosuppression, boost T-cell-derived antitumour cytokine secretion and enhance radiotherapy's effectiveness.[12 13] However, while many neoadjuvant studies focus on traditional CRT, only a few investigate the combination of nCRT with PD-1/PD-L1 antibodies for early rectal cancer. Notably, a Changhai Hospital phase II study presented at the 2022 ASCO meeting enrolled 23 patients with T1-3aN0-1 low rectal cancer.[14] They received concurrent sintilimab and LCRT, followed by sintilimab and chemotherapy. Among the patients, 43.2% achieved cCR, 20% achieved pCR, resulting in a combined CR rate of 52.2%. Anus preservation was at 95.5%, and grade 3–4 toxicities were 17.4%. These findings suggest that combining LCRT with PD-1 inhibitors could offer a highly effective and less toxic treatment option for challenging cases of early and low rectal cancer where surgical preservation is difficult.

Similar to Changhai study, many investigations into radiotherapy combined with PD-1/PD-L1 inhibitors for rectal cancer focus on LCRT. Comparably, SCRT combined with chemotherapy has also demonstrated favourable outcomes for patients with LARC in the Polish II, RAPIDO and STELLAR studies.[3 15 16] From the perspective of radiotherapy combined with immunotherapy, SCRT could be advantageous. Hypofractionated radiotherapy has shown less impact on peripheral blood lymphocytes in patients with locally advanced pancreatic cancer.[17] It can also curb myeloid-derived suppressor cell recruitment to tumours, decrease PD-L1 expression on tumour surfaces and yield improved tumour growth inhibition rates compared with conventional fractionation in mice,[18] even displaying abscopal effects when paired with immunotherapy.[19] These above findings provide a theoretical basis for the clinical trials of SCRT combined with PD-1/PD-L1 antibody and suggest the prospect of their application.

While there is a scarcity of reports on SCRT combined with immunotherapy for early rectal cancer, the TORCH study conducted by Fudan University adopted the TNT model that showcased promising tumour regression efficacy with SCRT and PD-1 inhibitor in patients with LARC.[20 21] The results remarkably showed that 60.4% patients achieved CR, 60.7% achieved pCR, 78.6% reached major pathological response and the organ preservation rate was 88.9%.[22] Furthermore, two additional SCRT-based trials also demonstrated substantial pCR rates in patients with LARC.[23 24] These prospective studies collectively suggest that integrating CRT, chemotherapy and PD-1/PD-L1 antibodies for patients with LARC could potentially yield higher tumour regression and CR rates compared with traditional chemoradiotherapy. Therefore, this approach holds promise for achieving favourable tumour responses and organ preservation rates when applied to low-lying early rectal cancer.

Based on the above findings and our previous TORCH study, the treatment approach of combining PD-1 monotherapy with SCRT and chemotherapy is expected to remarkably improve tumour regression and CR rate, and provide more options for patients with early rectal cancer to achieve organ preservation. Moreover, about 95% of patients with rectal cancer have the proficient mismatch repair (pMMR)/microsatellite-stable (MSS) type, which is not sensitive to PD-1 inhibitor alone. Therefore, we are conducting a phase II trial of the combination of SCRT, capecitabine plus oxaliplatin (CAPOX) and toripalimab for neoadjuvant therapy in pMMR/MSS, early and low rectal cancer, with the expectation to explore the efficacy and safety of this combination therapy.

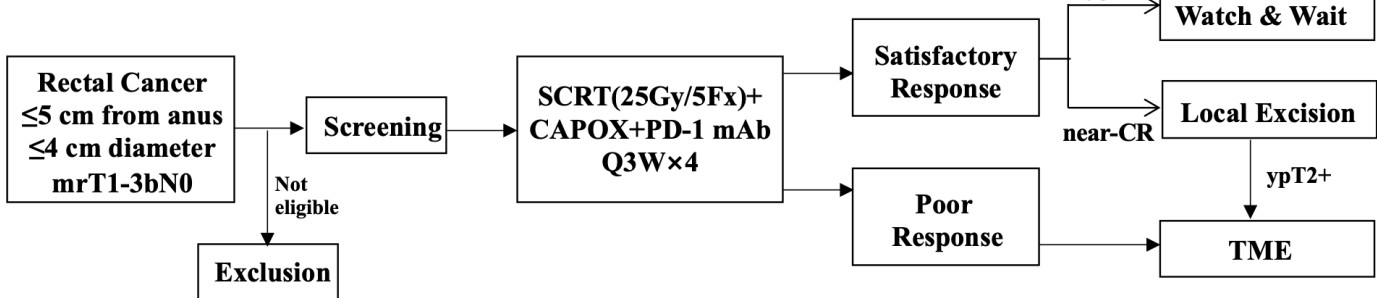

**Figure 1** Flow chart of the TORCH-E study. CAPOX, capecitabine plus oxaliplatin; PD-1:programmed cell death-1; cCR: clinical complete response; CR: clinical response; mAb, monoclonal antibody; SCRT, short-course radiotherapy; TME, total mesorectal excision.

## METHODS AND ANALYSIS

### Patient and public involvement

Neither patients nor the public will be involved in the design, recruitment, assessment and conduct of this study. The results will be disseminated via peer-reviewed scientific journals and conference presentations rather than specifically notified to a single patient.

### Protocol version

Protocol V.2.0, modified on 14 August 2023.

### Study design

The TORCH-E study is a multicentre, prospective phase II trial of short-term radiotherapy combined with chemotherapy and PD-1 inhibitor in early low rectal cancer. The enrolled patients will receive short-term radiotherapy (25 Gy/5 Fx), followed by four cycles of CAPOX chemotherapy and PD-1 antibody (toripalimab) and finally receive TME surgery or local excision (TEM). The CR (pCR plus cCR) rate, organ preservation rate, long-term prognosis and adverse effects will be analysed. The scheme of study progress was shown in figure 1.

### Trial organisation

This trial is investigator initiated by the Department of Radiation Oncology, Fudan University Shanghai Cancer Center (FUSCC). Five other cancer centres in China are involved in the study, including Renji Hospital, Shanghai Changzheng Hospital, Fujian Cancer Hospital, the Second Affiliated Hospital Zhejiang University School of Medicine and the Second Affiliated Hospital of Soochow University. Before enrolment of the first patient, the study protocol was approved by the Ethics Committee of all these research centres and all patients provided written informed consent before recruitment. Shanghai Junshi Biomedical Technology provides PD-1 antibody toripalimab free of charge and purchases liability insurance for clinical trial subjects. This trial was initiated in December 2022 and is currently recruiting patients in all of the participating institutions above.

### Study population

Inclusion in the TORCH-E trial requires that certain tumour and patient criteria are met. The inclusion and exclusion criteria are presented in table 1.

### Treatment

Patients will first receive a total dose of 25 Gy to the pelvic PTV delivered in fractions of 5 Gy. Almost 2 weeks after the last day of radiotherapy, four cycles of CAPOX chemotherapy and PD-1 antibody (toripalimab) treatment will be performed. The CAPOX regimen consists of capecitabine and oxaliplatin and is repeated every 3 weeks. Oxaliplatin ($130\,mg/m^2$) and toripalimab (240 mg) will be administered intravenously on day 1. Capecitabine is administered orally at $1000\,mg/m^2$ two times per day for 14 consecutive days (3 weeks per cycle). After the last cycle neoadjuvant treatment, response evaluation is possible to be performed. The multidisciplinary team will determine whether to proceed with local excision (TEM) or TME surgery based on the response evaluation. In cases where high-risk factors are identified after TEM, supplementary TME surgery is considered. Patients who achieve cCR are eligible for the W&W strategy.

The treatment termination criteria will be as follows: (1) disease progression during treatment; (2) encounter intolerable toxicity; (3) need emergency surgical resection due to intestinal obstruction, intestinal perforation, intestinal bleeding, etc; (4) request to withdraw from the cohort of this study due to various reasons; (5) accompanied by other non-tumour diseases make the patient unable to accept this treatment plan; (6) cannot complete the study plan due to various reasons.

### Study endpoints

The primary endpoint is the CR rate, including rate of pCR and cCR. Specifically, it includes the following three situations: (1) ypT0 for patients with local excisions; (2) ypT0N0/pCR confirmed after TME surgery; (3) cCR assessment for W&W patients. The cCR status is defined as undetectable tumour signs after neoadjuvant therapy through clinical examinations, including MRI of the pelvis, endoscopy and digital rectal examination (DRE) according to the criteria set by Maas et al.[25] The definition of a cCR is (1) substantial downsizing with no residual

**Table 1** Inclusion and exclusion criteria

| Inclusion criteria | Exclusion criteria |
| --- | --- |
| 1. Age 18–70 years old, female and male.<br>2. Pathological confirmed adenocarcinoma.<br>3. Clinical stage T1-3bN0, tumour maximum diameter less than 4 cm.<br>4. The distance from anal verge less than 5 cm.<br>5. Without distance metastases.<br>6. Karnofsky Performance Status (KPS) ≥70.<br>7. With good compliance.<br>8. Microsatellite repair status is MSS/pMMR.<br>9. Without previous anticancer therapy or immunotherapy.<br>10. Signed the informed consent. | 1. Pregnancy or breastfeeding women.<br>2. Pathological confirmed signet ring cell carcinoma.<br>3. Clinical stage T1N0 and can be resected locally.<br>4. History of other malignancies within 5 years.<br>5. Known history of severe neurological or mental illness.<br>6. Serious medical illness, such as severe mental disorders, cardiac disease, uncontrolled infection.<br>7. Immunodeficiency disease or long-term using of immunosuppressive agents.<br>8. Uncontrolled infection which needs systemic therapy.<br>9. Baseline blood and biochemical indicators do not meet the following criteria: neutrophils ≥1.5×10^9/L, Hemoglobin (Hb) ≥90 g/L, Platelet (PLT) ≥100×10^9/L, Alanine aminotransferase(ALT)/Aspartate Transaminase(AST)≤2.5 ULN, Creatinine (Cr) ≤1 ULN.<br>10. Dihydropyrimidine dehydrogenase (DPD) deficiency.<br>11. Allergic to any component of the therapy. |

MSS, microsatellite-stable; pMMR, proficient mismatch repair.

tumour or residual fibrosis only (with low signal on high b-value diffusion weighted imaging (DWI), if available). Residual wall thickening due to oedema only was also an indication for a possible cCR; (2) no suspicious lymph nodes on MRI; (3) no residual tumour at endoscopy or only a small residual erythematous ulcer or scar; (4) negative biopsies from the scar, ulcer or former tumour location; and (5) no palpable tumour, when initially palpable with DRE. If patients did not meet all of these criteria, they were regarded as incomplete responders. The pCR (ypT0) of local excision (LE) specimen is designated by no visible tumour cells,[8] and the pCR of TME specimen is defined as the absence of tumour cells at the primary site and regional lymph nodes. The secondary endpoints include the organ preservation rate, 3-year local recurrence-free survival (LRFS) rate, 3-year disease-free survival (DFS) rate, 3-year overall survival (OS) rate, grade 3–4 adverse effects rate and patients' QoL.

## Assessment

Regular examinations at the following time points will be performed: baseline, before the second PD-1 monoclonal antibody, after the last cycle of neoadjuvant treatment but prior to surgery and at every visit during the follow-up. Patients will undergo response evaluation with MRI of the pelvis, CT of chest and abdomen, DRE, blood routine examination, liver and renal function tests, myocardial enzymogram, thyroid hormone, adrenal hormone and serum tumour markers.

Radiological assessment will be according to the Response Evaluation Criteria in Solid Tumours[26] and the immune Response Evaluation Criteria in Solid Tumours.[27] The pathological evaluation is mainly used to evaluate the degree of tumour regression on the gross resected specimen. Pathological tumour regression grade will be evaluated according to the eighth American Joint Committee on Cancer Staging Manual by two independent pathologists.

Adverse effects will be assessed according to the Common Terminology Criteria for Adverse Event V.5.0, patients' QoL will be evaluated using the EORTC QLQ-C30(Europe Organization for Research and Treatment of Cancer,Quality of Life Questionnaire-C30) and EORTC QLQ-CR29 scales. Additionally, the evaluation of anorectal and bowel function will be based on the Wexner score and LARS score.[28 29]

### Sample size calculation and statistical analysis

This study is a prospective, single-arm, phase II clinical study. The primary endpoint is the CR rate. According to the previous reports, the reference CR rate is about 45% (null hypothesis, p0) and we assumed that the CR rate in this study can be increased to 70% (alternative hypothesis, p1). With the statistical power of 0.8 (β=0.2), the type I error rate of 0.05 (α=0.05), approximately 30 patients will be required in this study. Considering the dropout rate of 10%, the sample size in this study will be 34 cases.

The primary endpoint of TORCH-E study is the CR rate, including rate of pCR and cCR. The OS will be calculated from the date of randomisation to the date of death or the last follow-up. Survivals will be estimated by using the Kaplan-Meier method, including OS, DFS and LRFS. Endpoints related to the rate, including the cCR rate, pCR rate, organ preservation rate, toxicity occurrence, will be summarised with the frequency. The level of significance is p<0.05. All data analyses were performed using SPSS V.26.0.

### Follow-up

The follow-up will be carried out every 3 months for at least 3 years postoperatively. The follow-up contents include the physical examination, tumour marker

examination (carcinoembryonic antigen (CEA), carbohydrate antigen199 (CA199), alpha fetoprotein (AFP), CA724, CA242, CA50) and radiological examinations. MRI and CT scans will be performed alternately every 3 months. Colon endoscopy, late adverse effects collection, QoL assessment using EORTC questionnaires QLQ-C30 and QLQ-CR29, anorectal and bowel function using Wexner score and LARS score will be performed every 6 months. Colonoscopy, late adverse effects collection, QoL assessment (EORTC QLQ-C30/CR29), Wexner score and LARS score will be performed every 6 months. The events of local recurrence, distant metastasis and death will be recorded in detail.

## Data collection, management and monitoring

The data collection and management will be achieved by researchers. The participants in the study and their personal data will be kept confidential. All participants will be assigned a unique number, all records will be affiliated with this number and all study-related data will be anonymous. Finally, the clinical data will be analysed by the researchers. The results will be submitted for peer-reviewed publication and conference presentations.

## ETHICS AND DISSEMINATION

This study was conducted in compliance with Declaration of Helsinki principles. FUSCC is the sponsor, who takes overall responsibility for the initiation and management of this trial. Additionally, FUSCC is in charge of study design, data collection, analysis, interpretation and reporting, playing a role in ensuring ethical considerations, patient safety and adherence to regulatory guidelines throughout the entire research process. All procedures involving human subjects were approved by the Ethics Committee of FUSCC (Approval Number: 2209261–18). All patients signed informed consent forms before recruitment. The corresponding results will be disseminated via peer-reviewed journals and conference presentations. The participants in the study and their personal data during the study will be kept confidential in all study-related documents and publications.

**Author affiliations**
[1]Department of Radiation Oncology, Fudan University Shanghai Cancer Center, Shanghai, China
[2]Department of Oncology, Shanghai Medical College of Fudan University, Shanghai, China
[3]Department of Colorectal Surgery, Fudan University Shanghai Cancer Center, Shanghai, China

**Acknowledgements** We would like to thank Shanghai Junshi Biomedical Technology for their drug support and having purchased liability insurance for clinical trial subjects.

**Contributors** FX, ZZ and XL designed the clinical study. YC, Yaqi Wang, MZ, RW, WY, SZ and FX made substantial contributions to the organisation of this study. HZ, JW, LS, Yan Wang, XL, SC, ZZ and FX were responsible for patient recruitment. YC and FX contributed to the manuscript drafting and revising. All authors gave final approval the version of the protocol, and it was also approved by local researchers of the participating centres.

**Funding** This study was supported by Shanghai Junshi Biomedical Technology Co., Ltd. (Project protocol No. JS001-ISS-C0402), the National Natural Science Foundation of China (Grant No. 82102978) and the National Key R&D Program of China (Grant No.2022YFC2503700, 2022YFC2503702).

**Competing interests** None declared.

**Patient and public involvement** Patients and/or the public were not involved in the design, or conduct, or reporting, or dissemination plans of this research.

**Patient consent for publication** Not applicable.

**Provenance and peer review** Not commissioned; externally peer reviewed.

**ORCID iDs**
Yajie Chen http://orcid.org/0009-0006-9233-2572
Hui Zhang http://orcid.org/0000-0001-8803-2696
Juefeng Wan http://orcid.org/0000-0001-5361-1663

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
