## [Reviewer comments · BMJ Open]

ARTICLE DETAILS

TITLE (PROVISIONAL)	Short-course radiotherapy combined with chemotherapy and PD-1 inhibitor in low-lying early rectal cancer: study protocol for a single-arm, multi-center, prospective, phase II trial (TORCH-E)
AUTHORS	Chen, Yajie; Wang, Yaqi; Zhang, Hui; Wan, Juefeng; Shen, Lijun; Wang, Yan; Zhou, Menglong; Wu, Ruiyan; Yang, Wang; Zhou, Shujuan; Cai, Sanjun; Li, Xinxiang; Zhang, Zhen; Xia, Fan

VERSION 1 – REVIEW

REVIEWER	Savage, Joshua University of Birmingham, Cancer Research UK Clinical Trials Unit, Institute of Cancer and Genomic Sciences
REVIEW RETURNED	11-Jul-2023

GENERAL COMMENTS	This is a well-written and referenced manuscript with a detailed background and discussion for this trial of radiotherapy combined with chemolO in early rectal cancer. However, despite the SPIRIT checklist being completed, many of the criteria are not provided as described so these must be addressed in a revised manuscript, if it is to be accepted for publication. The SPIRIT checklist should indicate where the information is provided in the manuscript, so providing in the checklist or “Available on request” is not acceptable. • No mention of the version and date of the protocol that this manuscript is based on• Name and contact information for the trial sponsor, and details of the role of the sponsor and funder in the study conduct are not provided despite stating as such in SPIRIT checklist.• Study strengths and limitations should be kept factual; wording should be tempered “this study is of great clinical significance” as this is a bold claim by the authors• Typo in Abstract Background “adopted” should read “adopted”• Please define MPR• No mention of data management or statistical methods aside from the sample size calculation.• No mention of trial governance (management group, steering committee, monitoring committee), or audit. If this was not in place for the study or was not applicable this should be stated and justified.• No mention of patient anonymity and confidentiality, and how this is maintained.• Although “Availability of data and materials” is covered there is no detail on dissemination of the trial results to participants/public and healthcare professionals ; this should be detailed with plans, if any, for granting public access to the full protocol, participant-level
--

	dataset, and statistical code (and software used for statistical analysis). • The authors may consider including a model informed consent form and supporting documents in the Appendices.
--	---

REVIEWER	parasnis, amit Deenanath Mangeshkar Hospital and Research Centre, Oncosurgery
REVIEW RETURNED	13-Jul-2023

GENERAL COMMENTS	Nil
-----

REVIEWER	Zhang, Baoxia CSPC ZhongQi Pharmaceutical Technology (Shijiazhuang) Co., Ltd.
REVIEW RETURNED	18-Jul-2023

GENERAL COMMENTS	(1) Please complete a thorough proofread of the manuscript and correct any spelling and grammar errors that you identify. If possible, have the text modified by an English native speaker or enlist the help of a professional copy-editing service. (2) There is a great deal of detail about the treatment status of locally advanced rectal cancer (LARC) in the Introduction, would you please add more description of the unmet clinical needs of proficient mismatch repair (pMMR)/ microsatellite-stable (MSS) early and low rectal cancer? (3) In the paragraph of Treatment, it should be clear how many weeks are in a cycle. (4) What are the criteria for the treatment termination? (5) What are the definitions of the complete response and the incomplete response? (6) Please add the content of the statistical analysis method to the manuscript. (7) In the paragraph of Follow-up, which tumor marker will be examined in the trial? (8) Why use short-term radiotherapy?
---

VERSION 1 – AUTHOR RESPONSE

Response to the reviewer #1

This is a well-written and referenced manuscript with a detailed background and discussion for this trial of radiotherapy combined with chemoIO in early rectal cancer. However, despite the SPIRIT checklist being completed, many of the criteria are not provided as described so these must be addressed in a revised manuscript, if it is to be accepted for publication. The SPIRIT checklist should indicate where the information is provided in the manuscript, so providing in the checklist or “Available on request” is not acceptable.

Response: We are thankful for this comment and have provided more information in the SPIRIT checklist as well as the corresponding position in the revised manuscript.

• No mention of the version and date of the protocol that this manuscript is based on

Response: Thanks for this comment. We have added the version and date of the protocol in the “Methods and analysis” section in the revised manuscript. Protocol version: Protocol V.2.0, modified on 15 August 2023.

- Name and contact information for the trial sponsor, and details of the role of the sponsor and funder in the study conduct are not provided despite stating as such in SPIRIT checklist.

Response: Thanks for this comment. We have added the detailed information of the sponsor and funder in the “Ethics and dissemination” and “Funding” sections in the revised manuscript. Fudan University Shanghai Cancer Center (FUSCC) is the sponsor, who takes overall responsibility for the initiation and management of this trial. And this study was funded by Shanghai Junshi Biomedical Technology Co., Ltd. (Project protocol No. JS001-ISS-C0402), the National Natural Science Foundation of China (Grant No. 82102978) and the National Key R&D Program of China (Grant No.2022YFC2503700, 2022YFC2503702).

- Study strengths and limitations should be kept factual; wording should be tempered “this study is of great clinical significance” as this is a bold claim by the authors

Response: We appreciate the comment and have made a revision in the ‘Strengths and limitations of this study’ section. Strengths and limitations of this study: (1) This clinical trial is the first prospective, multi-center, phase II study using short-course radiotherapy combined with PD-1 inhibitor and chemotherapy in early low-lying rectal cancer. (2) This study aims to investigate the impact of short-course radiotherapy on tumor reduction and its synergy with immunotherapy, differing from previous studies employing long-course approaches. (3) The participants who achieved cCR can adopt the W&W strategy or local excision to preserve the rectum. (4) Limitation: this is a phase II clinical study involving a small number of patients, a phase III trial with a larger sample size will be necessary to validate the effectiveness and safety.

- Typo in Abstract Background “adpoted” should read “adopted”

Response: We are sorry for our careless typos and have modified as suggested.

- Please define MPR

Response: We are grateful for this comment, and have defined MPR in the Introduction section in the revised manuscript. MPR stands for “Major Pathological Response.” MPR was defined as $\leq 10\%$ of residual viable tumor in the surgical specimen[1, 2]. This response is indicative of a favorable therapeutic outcome and often correlates with improved patient prognosis and treatment efficacy.

- No mention of data management or statistical methods aside from the sample size calculation.

Response: We appreciate this valuable comment and have added the data management information in “Data collection, management and monitoring” section and statistical analysis methods in “Sample size calculation and statistical analysis” section in the revised manuscript.

- No mention of trial governance (management group, steering committee, monitoring committee), or audit. If this was not in place for the study or was not applicable this should be stated and justified.

Response: We appreciate the reviewer’s valuable comment and have added the data management information in “Data collection, management and monitoring” section.

In our study, while we do not have a formal management group, steering committee, or monitoring committee in place, we have implemented a rigorous internal oversight system to ensure the ethical and scientific integrity of the trial. This includes regular internal reviews by experienced oncologists, clinical research staff, and biostatisticians who closely monitor the study’s progress, data quality, and adherence to the protocol. Our commitment to maintaining the highest standards of clinical conduct and data integrity is of utmost importance. As for external audits, while not explicitly mentioned, we recognize the significance of this aspect in ensuring robust study conduct. In line with best practices, we are open to external audits should they be required or deemed necessary by regulatory authorities or ethical committees.

- No mention of patient anonymity and confidentiality, and how this is maintained.

Response: Thanks for this comment, and we have added the information in “Data collection, management and monitoring” section and “Ethics and dissemination” section. Participants will be assured that their identity will remain confidential in all study-related documents and publications. The model informed consent contains comprehensive explanations to participants regarding the protection of their personal information and the anonymization of data during analysis and reporting form has been uploaded in the Appendices.

The following is the content of the patient's anonymity and confidentiality in the informed consent form: The participants in the study and their personal data during the study will be kept confidential. The medical files will be kept in a locked filing cabinet and will only be accessible to researchers. A number is used in the study to identify participant's information and laboratory test specimens. The participants will not be identified and only the researcher and research team members can check the number. In order to ensure that the research is carried out in accordance with the regulations, when necessary, the research sponsor, government management department or members of the ethics review committee can check the participants' personal data in the research unit according to the regulations. When the results of this study are published, no personal information about the participants will be disclosed.

- Although “Availability of data and materials” is covered there is no detail on dissemination of the trial results to participants/public and healthcare professionals; this should be detailed with plans, if any, for granting public access to the full protocol, participant-level dataset, and statistical code (and software used for statistical analysis).

Response: We appreciate this comment and have added related information in the revised manuscript as suggested. Trial results will be disseminated via peer reviewed scientific journals and conference presentations.

- The authors may consider including a model informed consent form and supporting documents in the Appendices.

Response: We agree with the reviewer's suggestions and have uploaded the model informed consent form in the Appendices.

Response to the reviewer #3

(1) Please complete a thorough proofread of the manuscript and correct any spelling and grammar errors that you identify. If possible, have the text modified by an English native speaker or enlist the help of a professional copy-editing service.

Response: Thank you for your valuable comment. We appreciate your thorough review and your emphasis on ensuring the quality of the manuscript's language. We have taken your suggestions seriously and have conducted a comprehensive proofread of the manuscript to rectify any identified spelling and grammar errors.

(2) There is a great deal of detail about the treatment status of locally advanced rectal cancer (LARC) in the Introduction, would you please add more description of the unmet clinical needs of proficient mismatch repair (pMMR)/ microsatellite-stable (MSS) early and low rectal cancer?

Response: Thank you for your insightful comment and rearranged the content in the introduction section. In response to your suggestion, we have elaborated on three key unmet clinical needs. Firstly, the increasing demand for sphincter preservation. For patients with low-lying pMMR/MSS early rectal cancer where local resection is not feasible, preserving the anus becomes crucial for maintaining their quality of life. Thus, the neoadjuvant therapy followed by local excision might be an alternative to transabdominal rectal resection for patients with early-stage low rectal cancer who are unfit for major surgery, or seek preservation of the rectum. Additionally, there is an immediate imperative to augment the efficacy of neoadjuvant therapy. Emerging research underscores the

potential of neoadjuvant treatments to elevate complete response rates in early-stage tumors[3, 4]. While neoadjuvant treatments have shown benefit, optimizing the effectiveness and tailoring them to individual patients' needs is essential. Moreover, addressing immune resistance in MSS rectal cancer presents another unmet challenge. The remarkable efficacy and survival benefit of PD-1/PD-L1 inhibitors have been widely demonstrated in the treatment of various tumors, including MSI-H (microsatellite instability-high) colorectal cancer. However, MSI-H CRC patients only account for 5%-20%, and the rest MSS patients showed poor response to ICIs monotherapy[5, 6]. Hence, it's crucial to investigate whether merging radiotherapy, chemotherapy, and immunotherapy can enhance neoadjuvant efficacy in pMMR/MSS early and low rectal cancer.

(3) In the paragraph of Treatment, it should be clear how many weeks are in a cycle.

Response: Thanks for this comment. The CAPOX regimen will be performed for 3 weeks per cycle, we have added it in the paragraph of Treatment.

(4) What are the criteria for the treatment termination?

Response: We are thankful for this comment and provided more information about the termination criteria in the "Study population" section. The treatment termination criteria will be as follows: (1) disease progression during treatment; (2) encounter intolerable toxicity; (3) need emergency surgical resection due to intestinal obstruction, intestinal perforation, intestinal bleeding, etc.; (4) request to withdraw from the cohort of this study due to various reasons; (5) accompanied by other non-tumor diseases make the patient unable to accept this treatment plan; (6) cannot complete the study plan due to various reasons.

(5) What are the definitions of the complete response and the incomplete response?

Response: Thanks for this comment. Complete response refers to the complete disappearance or disappearance of signs and symptoms of disease for more than a certain period of time, usually assessed at the end of treatment, including clinical complete response(cCR) and pathological complete response(pCR).

The cCR is defined as undetectable tumour signs after nCRT by clinical examinations, including magnetic resonance imaging, endoscopy, and digital rectal exam according to the criteria set by Maas et al in 2011[7]. The definition of a cCR is (1) substantial downsizing with no residual tumor or residual fibrosis only (with low signal on high b-value DWI, if available). Residual wall thickening due to edema only was also an indication for a possible cCR; (2) no suspicious lymph nodes on MRI; (3) no residual tumor at endoscopy or only a small residual erythematous ulcer or scar; (4) negative biopsies from the scar, ulcer, or former tumor location; and (5) no palpable tumor, when initially palpable with digital rectal examination. If patients did not meet all of these criteria, they were regarded as incomplete responders.

In the pathology study of the local excision (LE) specimen, the size of the lesion (mm), degree of differentiation of adenocarcinoma, T stage, presence of venous, lymphatic or perineural infiltration, and margins (mm) were recorded. The pCR (ypT0) of LE specimen is designated by no visible tumor cells[3]. The pathology study of the TME specimen recorded the size of the lesion, degree of differentiation, presence of venous, lymphatic or perineural infiltration, distal and circumferential margins in mm and N stage (total lymph nodes found and positive lymph nodes). The pCR of TME specimen is defined as the absence of tumor cells at the primary site and regional lymph nodes[3]. The specimen will be assessed by two independent pathologists. Pathological tumour regression grade (pTRG) [8] will be evaluated according to the 8th American Joint Committee on Cancer (AJCC) Staging Manual. The pTRG and pCR status will be evaluated by two independent pathologists. If their conclusions are inconsistent, they will be evaluated again by a third pathologist.

(6) Please add the content of the statistical analysis method to the manuscript.

Response: We appreciate this valuable comment and have added the statistical analysis method in "Sample size calculation and statistical analysis" section in the revised manuscript.

(7) In the paragraph of Follow-up, which tumor marker will be examined in the trial?

Response: Thanks for this comment. The tumor marker will be examined include CEA, CA199, AFP, CA724, CA242, CA50. We have added these specific tumor markers in the revised manuscript.

(8) Why use short-term radiotherapy?

Response: Thanks for this comment. The utilization of short-term radiotherapy in this study is based on several key considerations. Firstly, short-course radiotherapy (SCRT) aims to reduce the size of the tumor and potentially downsize it to a more manageable state before surgical intervention. This can facilitate a less invasive surgical procedure, preserving organ function and improving postoperative outcomes. SCRT and sequential chemotherapy are commonly used modes of neoadjuvant treatment for patients with locally advanced rectal cancer that can achieve pCR rates similar to those of long-course radiotherapy (LCRT)[9].

Secondly, SCRT can create a more favorable tumor microenvironment by inducing immunogenic cell death and releasing tumor antigens. This, in turn, can enhance the immune system's recognition of cancer cells and promote a more robust anti-tumor immune response when combined with immunotherapy agents. Preclinical studies have shown that hypofractionated SCRT inhibits the recruitment of myeloid-derived suppressor cells (MDSCs) into tumors, reduces the expression of PD-L1 on the tumor surface and achieves a tumor growth inhibition rate that is superior to that of conventional fractionation[10]. From the perspective of radiotherapy combined with immunotherapy, SCRT may be more advantageous. SCRT has less effect on the peripheral blood lymphocytes of patients, thus promoting the antitumor effect of the immune system [11]. By priming the immune system through localized radiotherapy and then enhancing its activity with immunotherapy, we aim to create a more comprehensive and effective therapeutic strategy.

Furthermore, in our previous study, we have explored the model of SCRT combined with PD-1 inhibitor in LARC patients and showed high tumour regression efficacy with a pCR rate of 60.7%[9]. We have reasons to believe that SCRT and immunochemotherapy is expected to remarkably improve tumour regression in low early rectal cancer patients.

In conclusion, the utilization of short-term radiotherapy in our study is motivated by its potential to achieve optimal tumor regression when followed by chemotherapy, to create a favorable immunomodulatory environment that favors synergy with immunotherapy, and to be supported by our previous promising data in LARC patients.

References

1. Forde PM, Chaft JE, Pardoll DM: Neoadjuvant PD-1 Blockade in Resectable Lung Cancer. *N Engl J Med* 2018, 379(9):e14.
2. Chalabi M, Fanchi LF, Dijkstra KK, Van den Berg JG, Aalbers AG, Sikorska K, Lopez-Yurda M, Grootsholten C, Beets GL, Snaebjornsson P et al: Neoadjuvant immunotherapy leads to pathological responses in MMR-proficient and MMR-deficient early-stage colon cancers. *Nat Med* 2020, 26(4):566-576.
3. Serra-Aracil X, Pericay C, Badia-Closa J, Golda T, Biondo S, Hernandez P, Targarona E, Borda-Arrizabalaga N, Reina A, Delgado S et al: Short-term outcomes of chemoradiotherapy and local excision versus total mesorectal excision in T2-T3ab,N0,M0 rectal cancer: a multicentre randomised, controlled, phase III trial (the TAU-TEM study). *Ann Oncol* 2023, 34(1):78-90.
4. Simon Parkinson Bach, Johannes de Wilt, Femke Peters, Karen-Lise Garm Spindler, Ane Appelt, Mark Teo, Victoria Homer, Natalie Abbott, Ian Geh, Stephan Korsgen et al: Can we save the rectum by watchful waiting or transanal surgery following (chemo)Radiotherapy versus total mesorectal excision for early rectal cancer? In: *ASCO, 2022. Chicago.*
5. Le DT, Uram JN, Wang H, Bartlett BR, Kemberling H, Eyring AD, Skora AD, Luber BS, Azad NS, Laheru D et al: PD-1 Blockade in Tumors with Mismatch-Repair Deficiency. *N Engl J Med* 2015, 372(26):2509-2520.
6. YL. Verschoor, J. van den Berg, G. Beets, K. Sikorska, A. Aalbers, A. van Lent, C Grootsholten, I.

Huibregtse, H. Marsman, Oosterling. S et al: Neoadjuvant nivolumab, ipilimumab, and celecoxib in MMR-proficient and MMR-deficient colon cancers: Final clinical analysis of the NICHE study. In: ASCO, 2022; Chicago.

7. Maas M, Beets-Tan RG, Lambregts DM, Lammering G, Nelemans PJ, Engelen SM, van Dam RM, Jansen RL, Sosef M, Leijtens JW et al: Wait-and-see policy for clinical complete responders after chemoradiation for rectal cancer. *J Clin Oncol* 2011, 29(35):4633-4640.

8. Ryan R, Gibbons D, Hyland JM, Treanor D, White A, Mulcahy HE, O'Donoghue DP, Moriarty M, Fennelly D, Sheahan K: Pathological response following long-course neoadjuvant chemoradiotherapy for locally advanced rectal cancer. *Histopathology* 2005, 47(2):141-146.

9. Wang Y, Shen L, Wan J, Zhang H, Wu R, Wang J, Wang Y, Xu Y, Cai S, Zhang Z et al: Neoadjuvant chemoradiotherapy combined with immunotherapy for locally advanced rectal cancer: A new era for anal preservation. *Front Immunol* 2022, 13:1067036.

10. Lan J, Li R, Yin LM, Deng L, Gui J, Chen BQ, Zhou L, Meng MB, Huang QR, Mo XM et al: Targeting Myeloid-derived Suppressor Cells and Programmed Death Ligand 1 Confers Therapeutic Advantage of Ablative Hypofractionated Radiation Therapy Compared With Conventional Fractionated Radiation Therapy. *Int J Radiat Oncol Biol Phys* 2018, 101(1):74-87.

11. Crocenzi T, Cottam B, Newell P, Wolf RF, Hansen PD, Hammill C, Solhjem MC, To YY, Greathouse A, Tormoen G et al: A hypofractionated radiation regimen avoids the lymphopenia associated with neoadjuvant chemoradiation therapy of borderline resectable and locally advanced pancreatic adenocarcinoma. *J Immunother Cancer* 2016, 4:45.

VERSION 2 – REVIEW

REVIEWER	Savage, Joshua University of Birmingham, Cancer Research UK Clinical Trials Unit, Institute of Cancer and Genomic Sciences
REVIEW RETURNED	30-Aug-2023

GENERAL COMMENTS	This is a much improved manuscript which addresses most of my previous comments, and those of the editor and other reviewers. However, there is still no mention of independent trial oversight, monitoring, or audit by an iDMC and/or steering committee within the manuscript itself; see SPIRIT checklist points 5d and 21a, both of which are currently listed as NA. The authors provided a thorough response to this comment, but this should be incorporated into the manuscript.
---